# Enhancement of Hydrogen Productions by Accelerating Electron-Transfers of Sulfur Defects in the CuS@CuGaS$_2$ Heterojunction Photocatalysts

**Namgyu Son, Jun Neoung Heo, Young-Sang Youn, Youngsoo Kim, Jeong Yeon Do *** and **Misook Kang ***

Department of Chemistry, College of Science, Yeungnam University, Gyeongsan, Gyeongbuk 38541, Korea; sng1107@naver.com (N.S.); hjn2521@naver.com (J.N.H.); ysyoun@yu.ac.kr (Y.-S.Y.); kimys6553@yu.ac.kr (Y.K.)

* Correspondence: daengi77@ynu.ac.kr (J.Y.D.); mskang@ynu.ac.kr (M.K.); Tel.: +82-53-810-3798 (J.Y.D.); +82-53-810-2363 (M.K.); Fax: +82-53-815-5412 (M.K.)

**Abstract:** CuS and CuGaS$_2$ heterojunction catalysts were used to improve hydrogen production performance by photo splitting of methanol aqueous solution in the visible region in this study. CuGaS$_2$, which is a chalcogenide structure, can form structural defects to promote separation of electrons and holes and improve visible light absorbing ability. The optimum catalytic activity of CuGaS$_2$ was investigated by varying the heterojunction ratio of CuGaS$_2$ with CuS. Physicochemical properties of CuS, CuGaS$_2$ and CuS@CuGaS$_2$ nanoparticles were confirmed by X-ray diffraction, ultraviolet visible spectroscopy, high-resolution transmission electron microscopy, scanning electron microscopy and energy dispersive X-ray spectroscopy. Compared with pure CuS, the hydrogen production performance of CuGaS$_2$ doped with Ga dopant was improved by methanol photolysis, and the photoactivity of the heterogeneous CuS@CuGaS$_2$ catalyst was increased remarkably. Moreover, the 0.5CuS@1.5CuGaS$_2$ catalyst produced 3250 μmol of hydrogen through photolysis of aqueous methanol solution under 10 h UV light irradiation. According to the intensity modulated photovoltage spectroscopy (IMVS) results, the high photoactivity of the CuS@CuGaS$_2$ catalyst is attributed to the inhibition of recombination between electron-hole pairs, accelerating electron-transfer by acting as a trap site at the interface between CuGaS$_2$ structural defects and the heterojunction.

**Keywords:** hydrogen production; methanol photo-splitting; heterojunction; CuS@CuGaS$_2$; electron-hole recombination

## 1. Introduction

Copper sulfide (Cu$_{2-x}$S, 0< x < 1), a non-toxic and conductive chalcogen compound, has been continuously noted for its excellent photoelectric behavior, potential thermal/electrical properties, and unique biomedical properties for decades, and much extensive research on Cu$_{2-x}$S micro/nano structures is still being actively conducted. In particular, micro/nanostructured Cu$_{2-x}$S with well-controlled shapes, sizes, structures and compositions have already been applied as photocatalytic materials [1], energy conversion materials [2], biosensing materials [3], and bioimaging materials [4] and have shown reasonable results. However, comprehensive reviews of the Cu$_{2-x}$S structure in-depth in applications are still lacking. Therefore, it is necessary to categorize new functions or orientations of Cu$_{2-x}$S-based nanocomposites and to develop and improve their essential elements for specific applications. Many researchers have already published a number of strategies for synthesizing 0D (dimension) , 1D, 2D, and 3D micro/nanostructures (including polyhedra) [5–7], and their efforts have made important progress in identifying Cu$_{2-x}$S micro/nanostructures. Furthermore, improved Cu$_{2-x}$S

composites with hollow structures or super-lattices could be extended to a variety of applications in terms of performance [8]. $Cu_{2-x}S$ belongs to the covellite mineral group with a hexagonal crystal structure, and belongs to the crystal group of P63/mmc. However, bond lengths and angles can be varied in various ways depending on the oxidation state of the copper or other anion exchanges instead of $S^{2-}$ present in the surroundings. For example, $Cu_{2-x}S$ is very different from $Cu_{2-x}O$ but shows a similar structure to $Cu_{2-x}Se$ (klockmannite) [9]. CuS compound is paramagnetic due to the $3d^9$ electron arrangement, and some studies have reported that all Cu atoms in CuS have an oxidation state of $Cu^+$ based on XPS results [10]. However, it can be attributed to $(Cu^+)_2Cu^{2+}S_2$ with both Cu(I) and Cu(II) in the XRD crystal structure results [11]. There are many applications of $Cu_{2-x}S$ as a photocatalyst, among which Saranya et al. suggested that the morphology of CuS was influenced by the reaction time and surfactant, and its photocatalytic activity for decolorization of methylene blue (MB) dye under visible-light irradiation was 87% [12]. However, CuS is mainly used in combination with other types of photocatalysts rather than alone [13,14]. On the other hand, substitution of $Cu_{2-x}S$ with other metal ions instead of Cu(II) can produce a complex crystal structure with varying performance. For example, $CuInS_2$ and $CuGaS_2$, which are called CIS or CIGS, are used as light absorbers in thin film solar cells; they absorb visible light in a wide area and are stable to light, unlike $Cu_{2-x}S$. In particular, Salak et al. reported that $CuM^{3+}S_2$, a chalcogen compound, formed Cu defects on tetrahedrons and facilitated the separation of electrons and holes, which could maintain photoactivity for a long time [15]. Yue et al. concluded that $CuInS_2$ was most sensitive at 500 nm with an optimal apparent quantum yield of 23.85% [16]. Han et al. found that with an increase in the reaction temperature, the excitonic absorption peaks and band gap emission peaks were systematically red-shifted, thus exhibiting a quantum confinement effect, and the $CuGaS_2$ quantum dots showed promising visible-light-driven photocatalytic activity during degradation of rhodamine 6G [17]. We have already confirmed in previous studies that the $CuS@CuInS_2:In_2S_3$ catalyst has the ability to decompose water to produce hydrogen with high efficiency. Especially, it was found that the $CuInS_2$ layer inserted between CuS and $In_2S_3$ acts as an electron-rich interface to accelerate the reduction of water in this layer [18]. Furthermore, we have also found in previous studies that Ga has excellent performance in decomposing methanol to produce hydrogen, and that the hydrogen reverse-spillover phenomenon of Ga has a great influence on the removal of hydrogen from methanol [19]. Despite many previous studies, the excellent photosensitivity of chalcogen compounds, including $Cu_{2-x}S$, in multifunctional complexes is broadly applicable to a wide range of photochemical reactions, so research on chalcogen compounds remains of interest. In particular, if chalcogen compounds are expected to perform well in the photoreaction for hydrogen production from water decomposition and their photostability is guaranteed for a long time, development of the chalcogen photocatalyst will be quite a desirable area of study for the next generation of environmentally friendly energy sources.

　　　Therefore, in this study, we applied two particles of CuS and $CuGaS_2$ as base catalysts and applied them to hydrogen production from water degradation by hetero-connecting them between two particles. The effect of the pure structure or heterojunction structure of the two particles on photoactivity was investigated. Five types of catalysts were prepared: CuS, $CuGaS_2$, $1.0CuS@1.0CuGaS_2$, $0.5CuS@1.5CuGaS_2$, and $1.5CuS@0.5CuGaS_2$. The molar ratios of the two particles at the heterojunction were $CuS:CuGaS_2 = 1:1$, 0.5:1.5, and 1.5:0.5, respectively, to determine which particles most influence catalytic activity.

## 2. Results and Discussion

*Characteristics of CuS, CuGaS$_2$ and CuS@CuGaS$_2$ Nanoparticles*

　　　The XRD patterns (A) and high-resolution TEM (HRTEM) images (B) are shown in Figure 1 to confirm the crystallinity of synthesized CuS, $CuGaS_2$ and heterojunction $CuS@CuGaS_2$ nanoparticles. The main XRD peaks of CuS were observed at $2\theta = 27.65°$ (101), 29.25° (102), 31.75° (103), 32.77° (006), 38.77° (105), 47.89° (110), 52.61° (108), 59.24 (116), 73.87° (208) and 78.96° (213) and were

assigned to the covellite CuS of the hexagonal crystal structure (P63 / mmc space group, JCPDS card No. 01-078-0876) [20]. On the other hand, the peaks at 31.74° and 46.11° correspond to $Cu_2S$ (JCPDS card no. 00-053-0522) of cubic crystal structure, meaning that the two structures are finely mixed [21]. The XRD patterns of the $CuGaS_2$ nanoparticle showed a $CuGaS_2$ peak with a tetragonal crystal structure (I-42d space group, JCPDS card no. 01-085-1574) [22], although CuS was mixed. The main XRD peaks of $CuGaS_2$ were assigned to $2\theta = 29.11°$ (112), 33.49° (200), 48.63° (204) and 57.20° (312). Meanwhile, the XRD pattern of the heterojunction $CuS@CuGaS_2$ nanoparticles was very similar to the XRD pattern of the $CuGaS_2$ corresponding to the shell, but there was a slight difference in the intensity and position of the peaks as the ratio of $CuS:CuGaS_2$ varied. In particular, in the $0.5CuS@1.5CuGaS_2$ sample, the peak corresponding to the (112) crystal plane migrated at a higher angle than $CuGaS_2$. This is probably due to the small ionic radius of $Ga^{3+}$ ions compared to $Cu^{2+}$ [23], and it is expected that lattice parameters will be reduced according to Bragg's law [24], $n\lambda = 2d \sin(\theta)$ (where n is the order of reflection; the wavelength of the X-rays, d = the distance between two layers of the crystals, and $\theta$ = the angle of the incident light). It can be expected that lattice defects are formed as compared with pure CuS or $CuGaS_2$, since $Ga^{2+}$ ions can be incorporated into the CuS lattice or the lattice gap in the process of heterojunction. This can act as an active site of the catalyst and enhance catalyst performance. Figure 1B) shows the high-resolution TEM (HRTEM) (a), the selected area electron diffraction (SAED) (b), and the elemental mapping image (c) of the heterojunction $0.5CuS@1.5CuGaS_2$ particles. This result not only shows the overall shape of the particles, but also explains the intrinsic crystal structure of the particles based on the lattice parameter values in relation to the XRD results. The $0.5CuS@1.5CuGaS_2$ particles are shown in polycrystalline form as aggregates of single crystals with different orientations, which are evidenced by the lattice images and the SAED patterns. In general, when a certain point in a SAED pattern is clearly marked, it signifies a single crystal, and when a continuous ring is drawn, it signifies a polycrystalline. Therefore, heterojunction $0.5CuS@1.5CuGaS_2$ particles appeared to be polycrystalline, and lattice patterns of CuS and $CuGaS_2$ were observed. We can expect CuS and $CuGaS_2$ to be bonded as shown by the difference in shading in the TEM image. A lattice corresponding to 0.288 nm (103 diffraction plane) and 0.305 nm (102 diffraction plane) of CuS was observed in the bright portion, and a lattice pattern of 0.266 nm (200 diffraction plane) and 0.188 nm (204 diffraction plane) of $CuGaS_2$ was observed in dark areas. In particular, the 200 diffraction plane of $CuGaS_2$ has a very distinct lattice pattern, and a clear diffraction spot was also observed in the SAED pattern. This is consistent with the XRD results of the heterojunction $0.5CuS@1.5CuGaS_2$ particles and demonstrates that the particle is a continuous polycrystalline structure with a ring pattern with partially defined diffraction spots. On the other hand, the element mapping (c) results show that the Cu, Ga and S elements are uniformly distributed in the $0.5CuS@1.5CuGaS_2$ particles, thus demonstrating the microstructure and composition of the heterogeneous bonded particles.

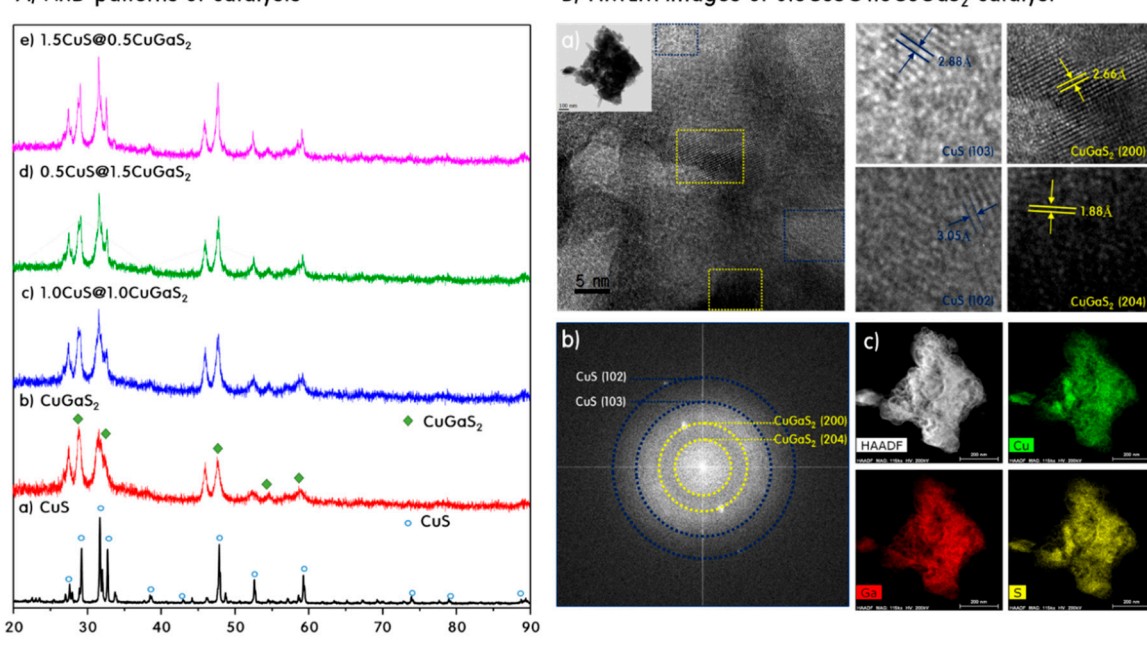

**Figure 1.** X-ray diffraction (XRD) patterns (**A**) and high-resolution transmission electron microscopy (HRTEM) images (**B**) of prepared samples.

Figure 2 shows the scanning electron microscope (SEM) image and energy-dispersive X-ray spectroscopy (EDS) analysis of synthesized CuS, CuGaS$_2$, and heterojunction CuS@CuGaS$_2$ nanoparticles, showing the composition of the components present on the surface of the particles. Figure 2A shows a SEM image of each sample, showing significant aggregation between the particles in the CuGaS$_2$ sample compared to CuS. On the other hand, the heterojunction CuS@CuGaS$_2$ sample inhibited the agglomeration of particles and the particle size became smaller. Figure 2B shows the EDS spectra of each sample and the atomic composition ratios are shown in the Table 1. Determination of the atomic ratio of the main metal species in the catalyst is very important because it relates to the density of the crystal lattice defects [25]. CuS and CuGaS$_2$, heterojunction CuS@CuGaS$_2$ particles show that Cu, Ga and S atomic components are precisely contained, and no other components are included. The atomic composition of pure CuS is close to the ideal stoichiometric mole fraction with a Cu:S ratio of 46.74:53.26. The composition of the CuGaS$_2$ sample was 31.46:27.26:41.27 with a Cu:Ga:S ratio slightly different from the stoichiometric ratio, but this can be predicted as a limitation of the EDS surface methodology. In addition, the heterojunction CuS@CuGaS$_2$ samples had a relatively low proportion of Ga. It is also expected that the difference in the sizes of Cu and Ga ions causes Ga to enter into the lattice of the crystal structure to reduce the amount of Ga exposed on the surface. Taking these factors into account, the overall molar ratio of Cu to Ga, S was almost quantitatively and reliably obtained.

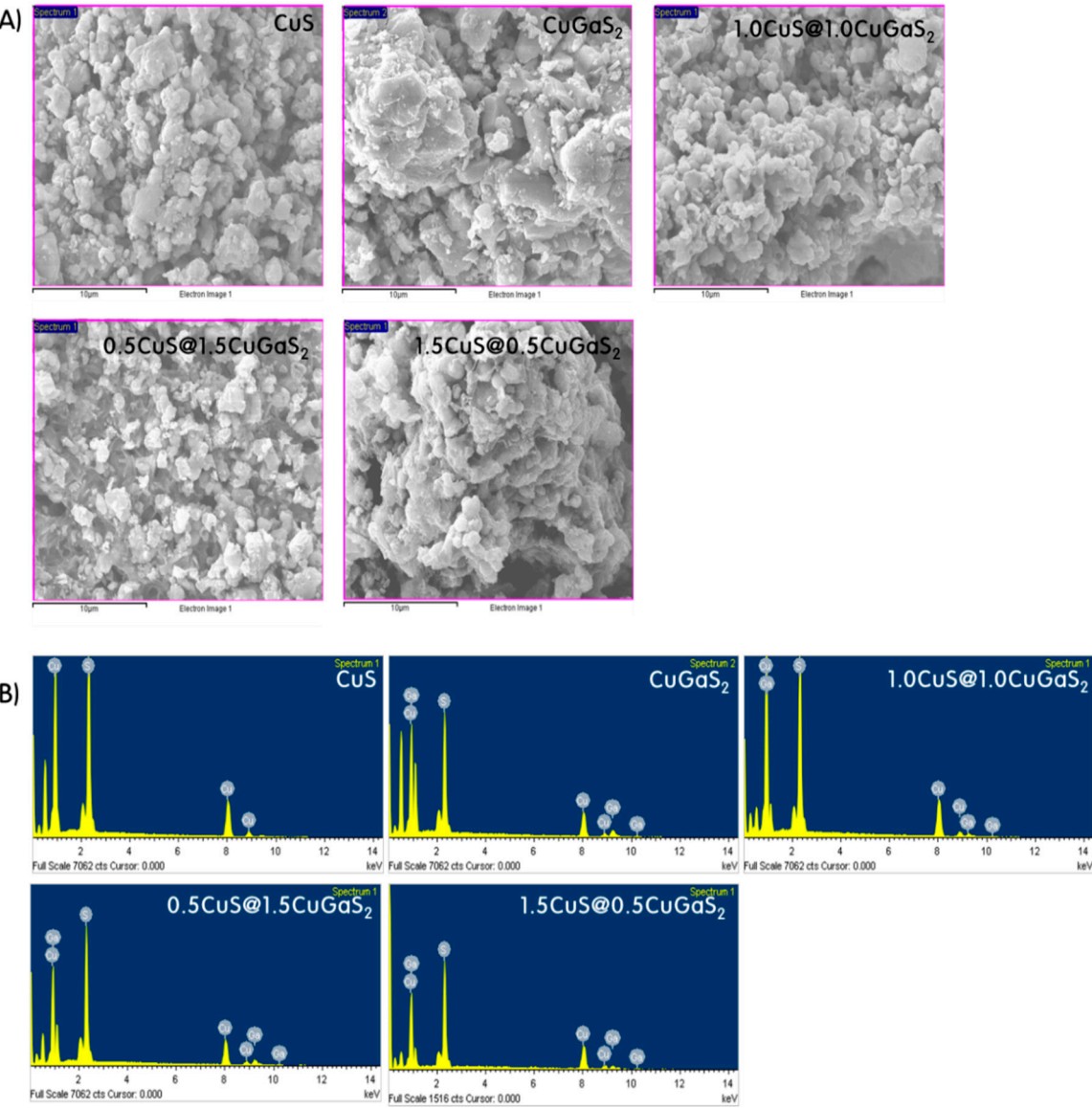

**Figure 2.** The SEM images (**A**) and energy-dispersive X-ray spectroscopy curves (**B**) of the CuS, $CuGaS_2$ and $CuS@CuGaS_2$ catalysts.

**Table 1.** Atomic compositions of the CuS, $CuGaS_2$ and $CuS@CuGaS_2$ catalysts determined by EDS.

| Catalysts/Elements | Cu | Ga | S |
|:---:|:---:|:---:|:---:|
| CuS | 46.74 | - | 53.26 |
| $CuGaS_2$ | 31.46 | 27.26 | 41.27 |
| $1.0CuS@1.0CuGaS_2$ | 32.99 | 23.06 | 43.95 |
| $0.5CuS@1.5CuGaS_2$ | 15.00 | 36.58 | 48.43 |
| $1.5CuS@0.5CuGaS_2$ | 41.05 | 18.31 | 40.64 |

Figure 3 shows the UV-visible reflectance spectra of the synthesized CuS, $CuGaS_2$ and heterojunction $CuS@CuGaS_2$ nanoparticles. UV-visible absorption spectroscopy is widely used to investigate microscopic changes caused by the chemical characteristics of the surface of the particles [26], and the optical band gap can be calculated through absorption spectra. Figure 3A clearly shows that the CuS, $CuGaS_2$ and heterojunction $CuS@CuGaS_2$ samples show absorption spectra in the region of 300~800 nm, and CuS in particular exhibited a pronounced absorption shoulder at about 620 nm. In the $CuGaS_2$ and $CuS@CuGaS_2$ samples, the apparent absorption shoulder peak

disappeared, but the overall absorption range was similar to CuS. These results suggest that CuS, $CuGaS_2$ and $CuS@CuGaS_2$ samples can be used as promising photocatalytic materials to absorb visible light. Based on the absorption spectrum, the band gap was calculated by the Tauc equation [27], $\alpha h\upsilon = A (h\upsilon = E_g)^n$. Where $h\upsilon$ is the photon energy, $\alpha$ is the absorption coefficient, A is the constant relative material, and n is the value that depends on the transitional nature (2 is direct allowed transition, 2/3 is direct suppressed transition, 2/3 is indirectly permissible transition). In addition, the energy band gap can be predicted from the wavelength extrapolated from the exciton peak called $\lambda_{1/2}$, or the point where the end of the absorption curve meets the $x$ axis. The band gaps of pure CuS samples and $CuGaS_2$ were 1.63 and 2.34 eV, respectively. This value is similar to the band gap reported in other papers [28]. The bandgap of the heterojunction $0.5CuS@1.5CuGaS_2$ particles was 2.32 eV. As $CuGaS_2$, which has a longer band gap, is bonded to CuS, $CuGaS_2$ first acts as a photosensitizer, and excited electrons can be transferred to CuS. If the bandgap is long, the recombination between the photogenerated electrons and the hole pair is delayed, and the photoactivity can be increased.

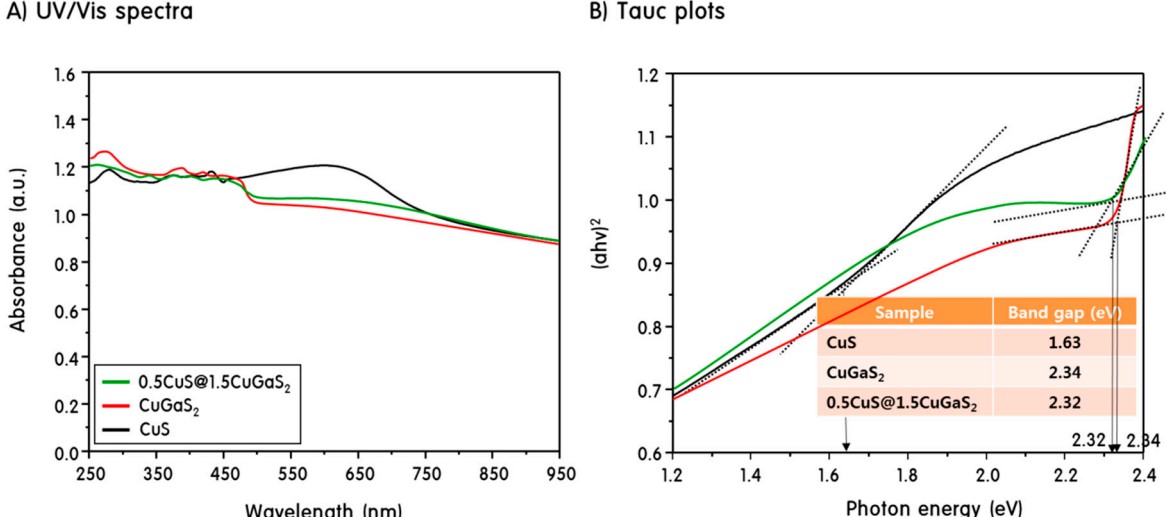

**Figure 3.** UV–visible spectroscopy curves (**A**) and Tauc plots (**B**) of CuS, $CuGaS_2$ and $CuS@CuGaS_2$ catalysts.

In general, the behavior of photogenerated carriers is closely related to photocatalytic activity. The generation, separation, transport and recombination of photogenerated electron-hole pairs have a great influence on photocatalytic activity [29]. Therefore, the photocurrent measurement and the photoluminescence measurement results are shown in Figure 4 in order to understand the separation efficiency and recombination characteristics of the photogenerated electrons and hole pairs. Figure 4A shows the results of measuring the photocurrent value when the light was irradiated by controlling the switch of the light source at intervals of 30 s. Electrons were excited by the irradiated light to generate excited electron and hole pairs. Their separation efficiency and mobility are closely related to the photocurrent value [30]. The photocurrent density values were increased in the order of CuS < $1.5CuS@0.5CuGaS_2$ < $CuGaS_2$ < $1.0CuS@1.0CuGaS_2$ < $0.5CuS@1.5CuGaS_2$. The pure CuS catalyst showed only a slight tendency for photocurrent density to drop momentarily when the light was turned off. In general photocurrent results, the photogenerated holes migrate to the catalyst surface and are captured or trapped by the reduced species in the electrolyte, and the electrons undergo backside contact through the catalyst, leading to an increase in the initial anodic photocurrent. Then, after the competitive separation of the electron and hole pairs and the equilibrium of the recombination, the photocurrent is kept constant, while the CuS temporarily decreases the photocurrent. This is presumably due to the fact that the traced holes on the catalyst surface are not captured or trapped by the reduced species in the electrolyte, but instead are competitively recombined with the electrons in

the conduction band of the catalyst [31]. On the other hand, the photocurrent density of CuGaS$_2$ and heterojunction CuS@CuGaS$_2$ catalysts was not only increased, but also showed excellent stability and reliability. It is believed that the addition of the Ga dopant induces more exciton formation and that structural or surface defects caused by heterojunction act as capture sites, or accelerate electron-transfer, resulting in more efficient photogenerated charge separation.

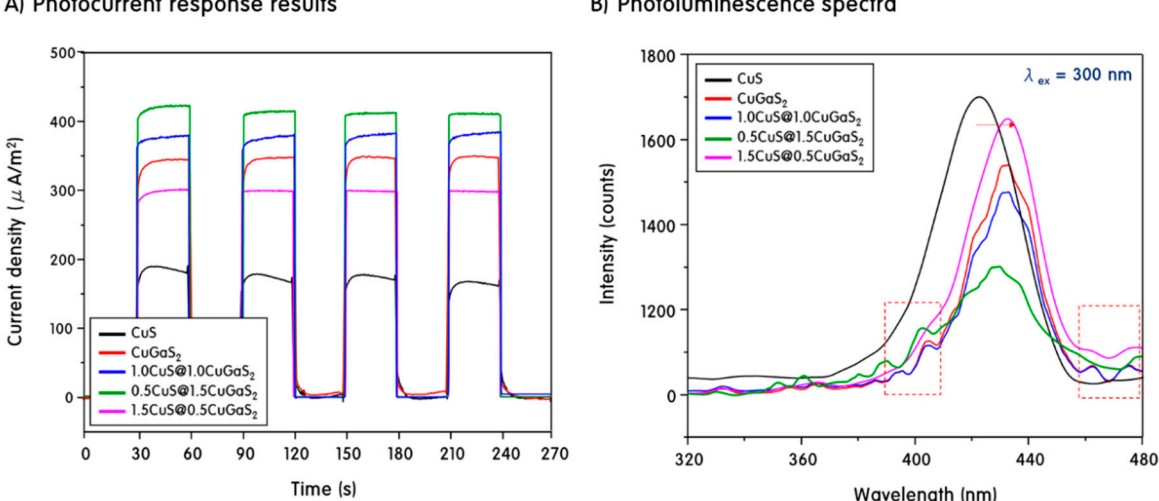

**Figure 4.** Photocurrent responses (**A**) and photoluminescence spectra (**B**) of CuS, CuGaS$_2$ and CuS@CuGaS$_2$ catalysts.

The transfer process of the photogenerated charge carrier is closely related to photoluminescence, and the photoluminescence measurement results are shown in Figure 4B. The position and intensity of the emission peak of photoluminescence (PL) differs depending on the kind of the catalyst and the vacancy [32]. Generally, as the intensity is increased, the recombination of the photogenerated electron and the hole pair is promoted and the photocatalytic activity is decreased [33]. When an excitation wavelength of 300 nm was irradiated, an emission peak was observed at about 420 nm in all of the CuS, CuGaS$_2$ and heterojunction CuS@CuGaS$_2$ nanoparticles. Especially, when Ga was added, the photoluminescence peak intensity of CuGaS$_2$ and the heterogeneous CuS@CuGaS$_2$ catalyst was slightly shifted to the long wavelength side. As the number of defect lattices generated in the Ga doping and heterojunction process increases, the electron trap site increases, which results in suppression of the recombination of photogenerated electrons and hole pairs. On the other hand, CuGaS$_2$ and heterojunction CuS@CuGaS$_2$ catalysts to which Ga was added, in comparison with pure CuS, had new emission peaks near 400 nm and 480 nm. According to Mehmood et al. [34], the blue emission peak at around 400 nm is due to a high energy defect due to the dopant. In addition, the electrons trapped by the dopant cannot generate excitons because they are bound by surface oxygen defects and other defects, thereby reducing the overall PL intensity. The blue-green emission peak at about 480 nm corresponds to the radiative transition of an electron to the deep donor level of the metal interstitials to an acceptor level of neutral V$_{metal}$ [35]. In contrast to photocurrent measurement, the photoluminescence intensity decreased in the order of CuS > 1.5CuS@0.5CuGaS$_2$ > CuGaS$_2$ > 1.0CuS@1.0CuGaS$_2$ > 0.5CuS@1.5CuGaS$_2$. As a result, the heterojunction 0.5CuS@1.5CuGaS$_2$ catalyst showed the lowest photoluminescence intensity, and the Ga doping and heterojunction structure can play an important role in increasing the light efficiency by slowing the recombination between exciton and hole pairs the most.

Based on the results of photoluminescence measurements, further intensity modulated photovoltage spectroscopy (IMVS) measurements were performed to compare the excited electron recombination lifetime for CuS, CuGaS$_2$ and heterojunction CuS@CuGaS$_2$ catalysts, and the results are shown in Figure 5A. IMVS is a useful method to study lifetime of electrons, which relates to

the electron recombination process [36]. The IMVS plot (A) of all samples showed a semicircular shape, and the larger the size of the semicircle in the IMVS results, the longer the recombination lifetime [37]. The electron recombination lifetime (B) was calculated from the IMVS plot using the equation [38] sr = 1/2 $\pi f_{min}$ , where $f_{min}$ is the frequency of the minimum imaginary component of the plot. As with the results of the photoluminescence measurement, the electron recombination lifetime increased in the order of: CuS < CuGaS$_2$ < CuS@CuGaS$_2$. The increased IMVS electron lifetime indicates that the residence time at the electron trap site increases. Figure 5B shows that CuGaS$_2$ increases the recombination lifetime due to the effect of Ga dopant compared to CuS, but shows the longest recombination life when the two catalysts are hetero-bonded. These results indicate that defects or catalyst surfaces formed during the heterojunction process generate more trap sites, and that the electron recombination lifetime is accelerated while accelerating electron-transfer, which may exert an excellent photocatalytic activity.

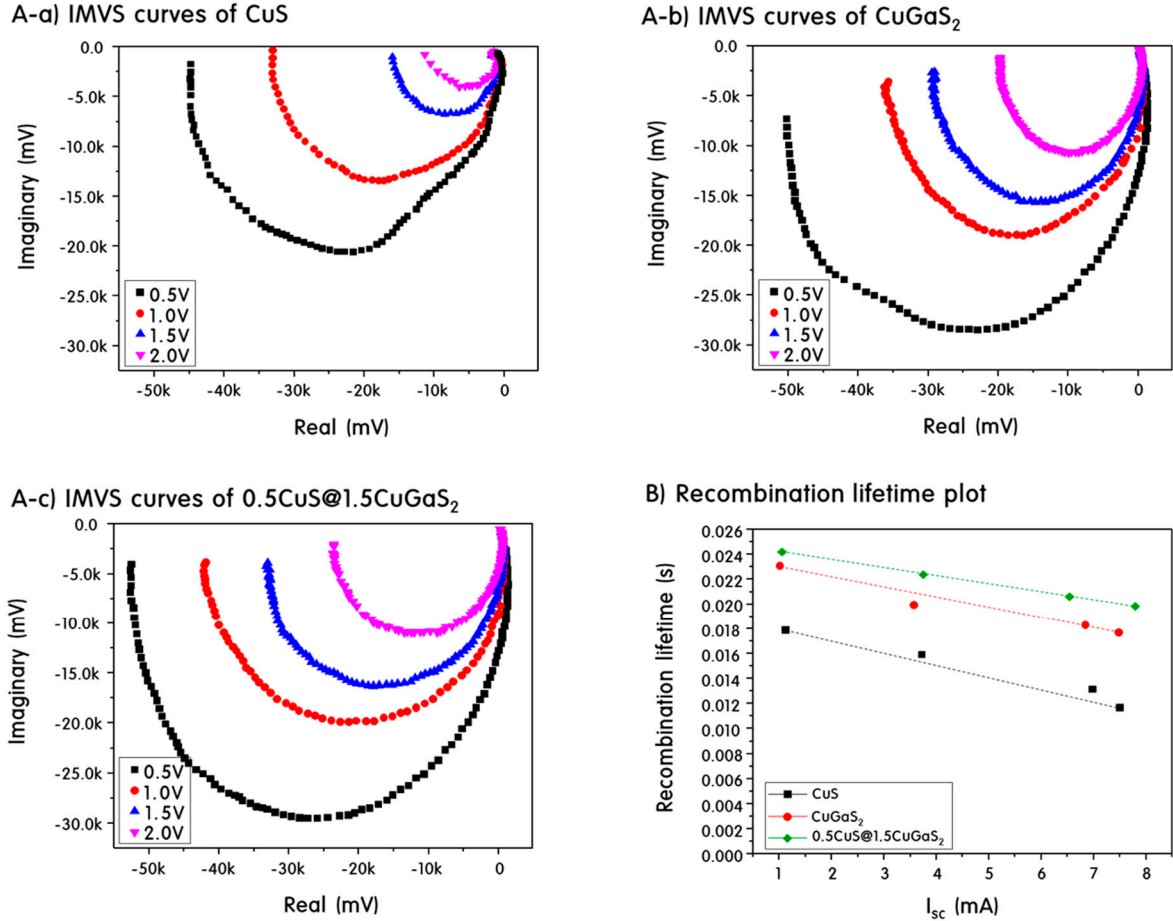

**Figure 5.** Intensity modulated photovoltage spectroscopy (IMVS) curves (**A**) and electron recombination lifetime (**B**) of CuS, CuGaS$_2$ and CuS@CuGaS$_2$ catalysts.

XPS was measured to investigate the chemical states of Cu, Ga and S ions in the surface state of the heterojunction 0.5CuS@1.5CuGaS$_2$ sample. The results are shown in Figure 6. In Cu atomic spectra, peaks corresponding to Cu 2p$_{3/2}$ and 2p$_{1/2}$ were observed at 934.27 and 953.71 eV. A typical satellite peak of the Cu$^{2+}$ oxidation state was observed at 942.78 and 963.06 eV, except for these two distinct peaks, indicating a defect in Cu$^{2+}$. This implies a vacancy in the surface state that occurs in vacancies or 0.5CuS@1.5CuGaS$_2$ heterojunction processes in skeletal distortions due to ion size differences due to Ga addition [39]. In the Ga atomic spectrum, peaks corresponding to 2p$_{3/2}$ and 2p$_{1/2}$ were observed at 1118.87 and 1145.80 eV, which corresponds to the Ga-S bond. On the other hand, three fitting curves were separated in the S atomic spectrum. The peak at low binding energy (162.00 eV) corresponds to

the Cu-S bond, and the peak at the high binding energy (165.28 eV) corresponds to the Ga-S bond [40]. Furthermore, the intermediate peak observed near 163.74 eV corresponds to sulfur vacancy, which is attributed to sulfur defects formed at the $CuGaS_2$ or heterojunction interface [41]. These XPS results demonstrate the presence of structural defects on the heterojunction $0.5CuS@1.5CuGaS_2$ catalyst surface and predict that the site can act as a reactive site.

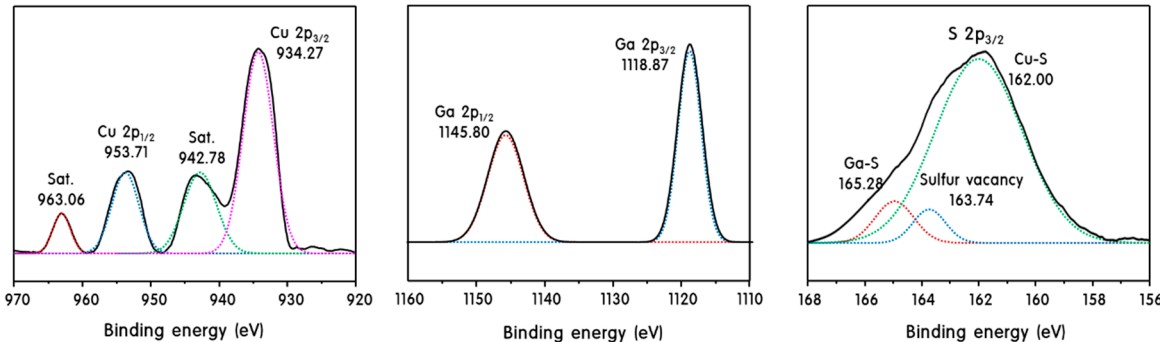

**Figure 6.** XPS spectra of heterojunction $0.5CuS@1.5CuGaS_2$ catalysts.

Figure 7 summarizes the evolution of hydrogen from the photo splitting of aqueous methanol solution to CuS, $CuGaS_2$ and $CuS@CuGaS_2$ catalysts. Figure 7A shows the amount of hydrogen generated under 365 nm UV light source conditions of the catalyst. The pure CuS catalyst showed little hydrogen evolution even after 10 h of reaction. On the other hand, the amount of hydrogen generation in the $CuGaS_2$ catalyst was remarkably increased, and the amount of hydrogen produced after the reaction for 10 h reached 3000 μmol. In particular, the $CuS@CuGaS_2$ and $0.5CuS@1.5CuGaS_2$ catalysts increased the amount of hydrogen generation further, reaching 3100 and 3250 μmol, respectively. Meanwhile, the heterojunction $1.5CuS@0.5CuGaS_2$ catalyst with a high CuS ratio produced less hydrogen than $CuGaS_2$. This is considered to be due to the fact that the portion exposed on the surface of the catalyst has a large amount of CuS and is less influenced by $CuGaS_2$. According to the previous studies [42], the photo splitting process of methanol aqueous solution follows the following reaction:

$$CH_3OH + H_2O \rightarrow CO_2 + 6H^+ + 6e^- \qquad (1)$$

$$6H^+ + 6e^- \rightarrow 3H_2 \qquad (2)$$

$$CH_3OH + H_2O \rightarrow CO_2 + 3H_2 \qquad (3)$$

According to the above photolytic decomposition method of methanol, hydrogen and carbon dioxide are produced, but in this study, carbon dioxide was not observed because it exists as a $CO_2$ ion in an aqueous solution. In addition, photolysis of methanol aqueous solution was further performed using a 150 W Xe lamp to confirm the catalytic activity in the visible region. The amount of hydrogen produced was reduced by about 1/20 compared to the UV light source, and the $0.5CuS@1.5CuGaS_2$ catalyst, which was heterogeneously bonded to the $CuGaS_2$ catalyst, produced about 200 μmol of hydrogen. From these results, we have confirmed that $CuGaS_2$ and the heterogeneous $CuS@CuGaS_2$ catalyst exhibit optical activity even in the visible region, albeit in a small amount compared to UV light sources.

From these results, we proposed an improved photocatalytic decomposition of aqueous solution of $CuS@CuGaS_2$ catalyst as shown in Scheme 1. The valence band, conduction band, and band gap values of CuS [18] and $CuGaS_2$ [43] have already been reported in other studies and based on this, energy potential diagrams are shown together. In the heterojunction $CuS@CuGaS_2$, the band gap of $CuGaS_2$ contains CuS, and $CuGaS_2$ first acts as a light-absorbing agent. Electrons are excited from the valence band to the conduction band by light irradiation, and electrons generated from $CuGaS_2$ can move to the conduction band of the adjacent CuS. This transfer is thermodynamically favorable

by band gap alignment, and the photogenerated electrons react with $H^+$ to produce $H_2$. At this time, $S^{2-}/S_x^{2-}$, which is a sacrifice material of the metal sulfide, captures holes by the following equation [44], and suppresses recombination between electron-hole pairs.

$$(1) \quad 2\,S^{2-} + 2\,h^+ \rightarrow S_2^{2-}$$
$$(2) \quad S^{2-} + 2h^+ \rightarrow S$$

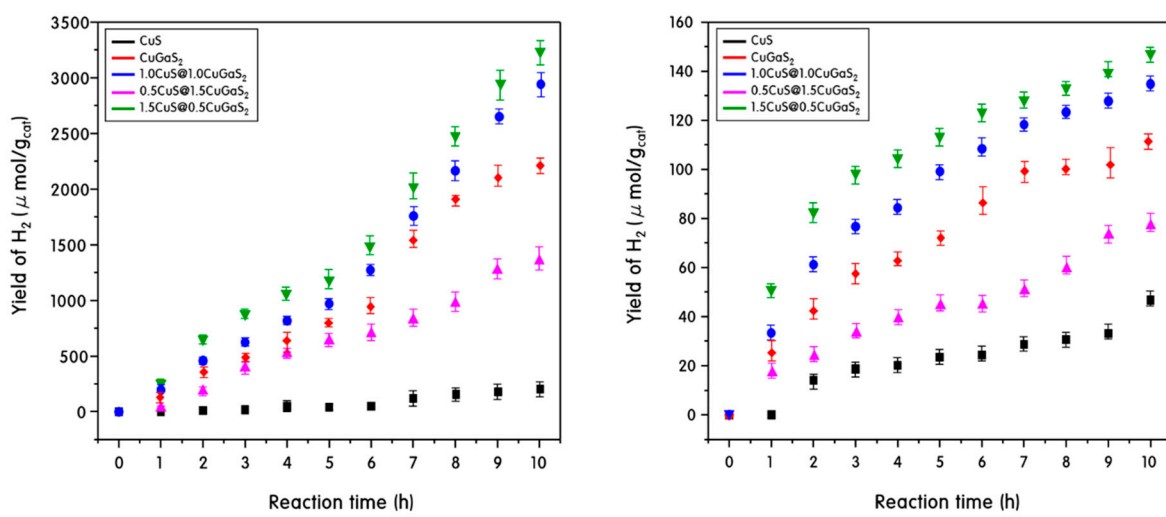

**Figure 7.** Evolution of $H_2$ for methanol aqueous solution photo-splitting under UV light source (**A**) and visible light source (**B**) for CuS, CuGaS$_2$ and CuS@CuGaS$_2$.

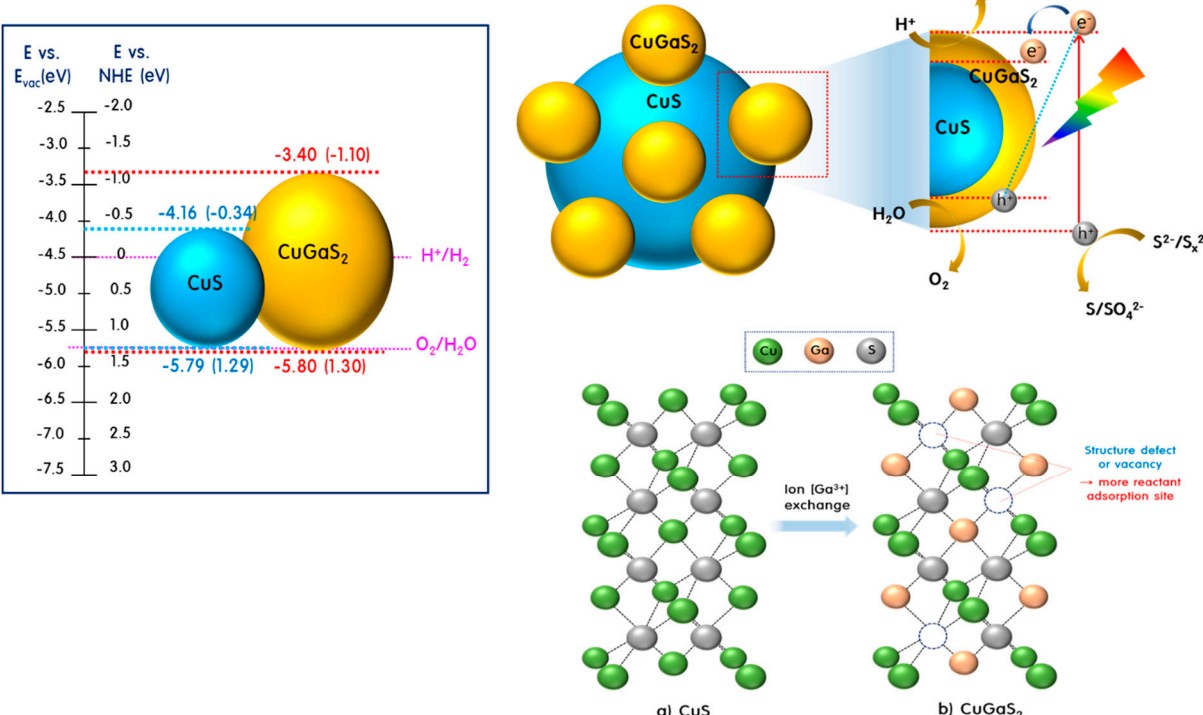

**Scheme 1.** The expected mechanism for methanol aqueous solution photo-splitting in photosystem with CuS@CuGaS$_2$ catalyst.

In addition, structural defects of the $CuGaS_2$ catalyst caused by the interface between the heterojunction $CuS@CuGaS_2$ catalyst and the $GaGaS_2$ catalyst may act as a trap site to accelerate hole transfer and prolong the electron lifetime. In $CuGaS_2$ structure, free electrons are gathered around $Ga^{3+}$ for charge balance based on CuS structure, and sulfur vacancies due to structural defects are formed. These sites provide a place where the reactants can be adsorbed better, resulting in more products, and ultimately an improvement in photocatalytic activity. Moreover, structural defects at the $CuS@CuGaS_2$ catalyst interface form quasi-continuous energy levels and reduce ohmic contact to induce ohmic contact [28]. This contact recombines the holes formed in VB of CuS and the excited electrons in CB of $CuGaS_2$, which ultimately promotes the efficiency of photocatalytic activity by promoting the separation efficiency of the photogenerated charge pair in the heterojunction $CuS@CuGaS_2$ complex catalyst.

## 3. Experimental

### 3.1. Preparation of CuS, CuGaS₂ and CuS@CuGaS₂ Nanoparticles

CuS and $CuGaS_2$ nanoparticles were prepared using a typical sol-gel synthesis method [45]. Copper (II) nitrate trihydrate ($Cu(NO_3)_2 \cdot 3H_2O$, 99.0%, Junsei Chemical, Tokyo, Japan), Gallium (III) nitrate hydrate ($Ga(NO_3)_3 \cdot xH_2O$, 99.9%, Alfa Aesar, Tewksbury, MA, USA) and Thiourea ($CH_4N_2S$, 98.0%, Junsei Chemical, Tokyo, Japan) were used as starting materials for Cu, Ga and S, respectively. First, in order to synthesize CuS, $Cu(NO_3)_2 \cdot 3H_2O$ and $CH_4N_2S$ were dissolved in ethylene glycol at a molar ratio of 1:2, mixed well and aged at 180 °C for 8 h. The resulting powder was treated at 400 °C for 4 h under an argon atmosphere to obtain black CuS nanoparticles. In the synthesis of $CuGaS_2$ particles, only the molar ratio of $Cu(NO_3)_2 \cdot 3H_2O$, $Ga(NO_3)_3 \cdot xH_2O$ and $CH_4N_2S$ was changed to 1:1.25:4 during the synthesis of CuS, respectively.

On the other hand, heterogeneous $CuS@CuGaS_2$ nanoparticles were obtained by the impregnation method [46] using prepared CuS and $CuGaS_2$. The synthesis procedure was as follows. The amounts of CuS and $CuGaS_2$ added were different. CuS was added to ethanol, and the mixture was stirred for 2 h, then $CuGaS_2$ was added and stirred sufficiently. The homogeneously stirred solution was separated into powdery samples by centrifugation and dried at 80 °C for 24 h. Thereafter, the resultant was again annealed at 200 °C for 2 h in order to remove impurities and increase the bonding strength, finally obtaining a heterogeneous $CuS@CuGaS_2$ catalyst.

### 3.2. Characterization of CuS, CuGaS₂ and CuS@CuGaS₂ Nanoparticles

X-ray diffraction (XRD, MPD, PANalytical, Almelo, The Netherlands) was used to analyze the crystal structure of the prepared CuS, $CuGaS_2$ and heterojuntion $CuS@CuGaS_2$ nanoparticles. XRD was measured at a 2θ angle of 20–100° using nickel-filtered $CuK\alpha$ (λ = 1.5056 Å) radiation (40 kV, 30 mA). The shape and size of the particles were confirmed using high-resolution transmission electron microscopy (TEM, H-7600, Hitachi, Tokyo, Japan) and scanning electron microscopy (SEM, S-4100, Hitachi, Tokyo, Japan). In addition, energy-dispersive X-ray spectroscope (EDS, EX-250, Horiba, Kyoto, Japan) analysis was used to identify the atomic composition of CuS, $CuGaS_2$, and heterogeneous $CuS@CuGaS_2$ nanoparticles.

The diffuse reflection spectra of the particles were obtained using a UV-Vis spectrophotometer (Neosys-2000, SCINCO, Daejeon, Korea). The recombination tendency between the photogenerated electron-hole pair ($e^-/h^+$) of the catalyst was determined using a photoluminescence spectroscopy (PL, FS-2, SCINCO, Daejeon, Korea) equipped with a 150 W continuous Xenon lamp light source.

In addition, photocurrent and intensity modulated photovoltage spectroscopy (IMVS) measurements were taken with a two-electrode system to confirm the behavior of the photogenerated charge carrier. The catalyst was coated on fluorine doped tin oxide (FTO) glass to form a cell, and a platinum wire was used as a counter electrode. The catalyst coated on the FTO glass with a certain unit area was used as the working electrode and photocurrent was measured by irradiating light at intervals

of 30 s. The IMVS measurement was also performed using a visible light source in a two-electrode system, and the recombination lifetime of electrons was confirmed through the measurement.

*3.3. Hydrogen Production by Photo Splitting of Methanol Aqueous Solution Using CuS, CuGaS$_2$ and Heterojunction CuS@CuGaS$_2$ Catalyst*

The photocatalytic decomposition of methanol aqueous solution was carried out using a liquid photoreactor prepared in our laboratory, which was reported in previous work [47]. First, the photocatalytic decomposition of methanol aqueous solution using a UV light source was performed using a pyrex reactor. 1.0 L of a mixed solution of 500 mL of methanol and 500 mL of distilled water was put into the reactor, and 0.5 g of the synthesized CuS, CuGaS$_2$ and CuS@CuGaS$_2$ powder was added. Light was irradiated using a UV-lamp ($3 \times 6$ cm$^{-2}$ = 18 W cm$^{-2}$, length 30 cm, diameter 2.0 cm, Shinan, Pochon, Korea) at a wavelength of 365 nm and the reaction was performed for a total of 10 h. The photocatalytic decomposition of methanol aqueous solution using a visible light source was performed in a quartz reactor using a 150 W Xe lamp. The resulting gas was analyzed by gas chromatography (GC, DS7200, Donam Company, Gwangju, Korea) equipped with a thermal conductivity detector (TCD). The following GC conditions were used: TCD detector, Carboxen-1000 column (Bruker, Billerica, MA, USA), and the injection, oven and detector temperatures of 423, 393 and 473 K, respectively.

## 4. Conclusions

We have synthesized CuS, CuGaS$_2$, and heterojunction CuS@CuGaS$_2$ catalysts for hydrogen production through methanol aqueous photo splitting. The interface between the heterojunction CuS@CuGaS$_2$ catalyst and the structural defect of CuGaS$_2$ formed by the addition of Ga$^{3+}$ to CuS acted as a trap site. This trap site accelerates the electron-transfer, indicating a high photocurrent density value in the photocurrent results and excellent charge separation efficiency. In addition, compared to pure CuS and CuGaS$_2$, as shown by photoluminescence and IMVS measurements, recombination between the excited electron-hole pairs in the heterojunction catalyst was suppressed, resulting in higher electron lifetime. As a result, the heterojunction CuS@CuGaS$_2$ catalyst produced a significant amount of hydrogen gas, up to 3250 and 200 μmol, through photo splitting of aqueous methanol solution under UV and visible light irradiation, showing a significant increase in photocatalytic activity.

**Author Contributions:** Conceptualization, M.K.; Data curation, N.S. and J.Y.D.; Formal analysis, N.S. and J.Y.D.; Investigation, J.N.H. and Y.K.; Methodology, J.N.H. and Y.-S.Y.; Supervision, M.K.; Writing—original draft, J.Y.D.; Writing—review & editing, M.K.

**Funding:** This work was supported by the National Research Foundation of Korea (NRF) grant funded by the Korean government (MSIT) (No. 2018R1A2B6004746).

**Conflicts of Interest:** The authors declare no conflict of interest.

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
