# Peer review of "Enhancement of Hydrogen Productions by Accelerating Electron-Transfers of Sulfur Defects in the CuS@CuGaS2 Heterojunction Photocatalysts"

_catalysts, doi:10.3390/catal9010041_

Round 1
Reviewer 1 Report
The present manuscript reports on CuS@CuGaS2 heterojunction which enhances hydrogen production by accelerating electron-transfers of sulfur defects. The paper is probably publishable, but needs major revision before any further consideration for publication. The main concerns include:
1. The Scheme 1 describing the mechanism is too general. There is no information about the location (the values) of VB and CB in CuS and CuGaS2 and corresponding band gap energy vaules, this is important to identify which type of heterojunction is present. Furthermore, the discussion about the mechanism of identified heterojunction between CuS and CuGaS2 should be extended.
2. The method of determination of band gap energies presented in Fig. 3b is not clear and does not exactly correspond to the cited reference (Fig. 2b in Ref.27).
3. There is no information about the wavelength range for photocatalytic experiment conducted under visible light irradiation. Was any cut-off filter used? Why has been monochromatic irradiation (365 nm) used in the case of UV irradiation?
Author Response
Thank you for your kind comments. We modified the manuscript according to your comments, and uploaded the specific answers as attachments.

Reviewer 2 Report
The paper by Son et al. reports interesting results on hydrogen productions by accelerating electron-transfers of sulfur defects in the CuS@CuGaS2 heterojunction photocatalysts The degree of innovation is good, as well as the potential impact. However, some amendments are necessary before publication. Especially, it is particularly important to assess the role of chalcogen vacancies in surface chemical reactivity, properly citing the state of the art. The presence of defects is able to transform pristine materials, such as graphene [1] and metal dichalcogenides [2], into selective catalysts
[1] Unveiling the Mechanisms Leading to H2 Production Promoted by Water Decomposition on Epitaxial Graphene at Room Temperature, ACS Nano 10 (2016) 4543.
[2] Tailoring the Surface Chemical Reactivity of Transition-Metal Dichalcogenide PtTe2 Crystals, Adv. Funct. Mater. 28 (2018) 1706504
Author Response

(The authors gave the same response as above.)

Round 2
Reviewer 1 Report
Thank you for your replies. I accept the manuscript in the present form.